# Optical Intracranial Self-Stimulation (oICSS): A New Behavioral Model for Studying Drug Reward and Aversion in Rodents

**DOI:** 10.3390/ijms25063455

**Published:** 2024-03-19

**Authors:** Rui Song, Omar Soler-Cedeño, Zheng-Xiong Xi

**Affiliations:** 1Beijing Key Laboratory of Neuropsychopharmacology, Beijing Institute of Pharmacology and Toxicology (BIPT), 27th Taiping Road, Beijing 100850, China; 2Addiction Biology Unit, Molecular Targets and Medications Discovery Branch, National Institute on Drug Abuse (NIDA), Intramural Research Program (IRP), Baltimore, MD 21224, USA; omar.soler-cedeno@nih.gov

**Keywords:** intracranial self-stimulation (ICSS), brain-stimulation reward (BSR), optogenetics, optical ICSS, reward, aversion, drugs of abuse

## Abstract

Brain-stimulation reward, also known as intracranial self-stimulation (ICSS), is a commonly used procedure for studying brain reward function and drug reward. In electrical ICSS (eICSS), an electrode is surgically implanted into the medial forebrain bundle (MFB) in the lateral hypothalamus or the ventral tegmental area (VTA) in the midbrain. Operant lever responding leads to the delivery of electrical pulse stimulation. The alteration in the stimulation frequency-lever response curve is used to evaluate the impact of pharmacological agents on brain reward function. If a test drug induces a leftward or upward shift in the eICSS response curve, it implies a reward-enhancing or abuse-like effect. Conversely, if a drug causes a rightward or downward shift in the functional response curve, it suggests a reward-attenuating or aversive effect. A significant drawback of eICSS is the lack of cellular selectivity in understanding the neural substrates underlying this behavior. Excitingly, recent advancements in optical ICSS (oICSS) have facilitated the development of at least three cell type-specific oICSS models—dopamine-, glutamate-, and GABA-dependent oICSS. In these new models, a comparable stimulation frequency-lever response curve has been established and employed to study the substrate-specific mechanisms underlying brain reward function and a drug’s rewarding versus aversive effects. In this review article, we summarize recent progress in this exciting research area. The findings in oICSS have not only increased our understanding of the neural mechanisms underlying drug reward and addiction but have also introduced a novel behavioral model in preclinical medication development for treating substance use disorders.

## 1. Introduction

Brain-stimulation reward (BSR) or intracranial self-stimulation (ICSS) is a classical experimental paradigm used to study the neural substrates underlying reward processes and motivated behavior [1,2,3,4]. BSR is the pleasurable experience induced by direct stimulation of brain reward regions, while ICSS is an operant behavior producing BSR. This technique can be traced back to the 1950s when Olds and Milner introduced the concept that electrical stimulation of certain brain regions could induce pleasurable experiences in rats, giving birth to the intriguing realm of ICSS [5]. Over time, researchers identified key structures supporting this behavior, including the lateral hypothalamus, the medial forebrain bundle (MFB), and the ventral tegmental area (VTA) in the midbrain [4,6,7,8,9,10].

The electrical ICSS (eICSS) procedure has since become a versatile tool with a multitude of applications. Firstly, it has been extensively utilized to identify the neural substrates and circuits responsible for brain reward function [1,5,10]. In this procedure, subjects are given a condition to operantly respond to electrical stimulation of a specific brain region using a box with two levers or nose pokes—one active contingent to electrical stimulation and another inactive without consequence upon responding. The number of active responses and the discrimination between active and inactive levers or nose pokes are crucial measures. If a subject makes more active responses to obtain stimulation, it suggests that the targeted brain region supports positive eICSS, indicating its involvement in positively reinforcing effects [6,8,11]. Thus, the eICSS procedures have provided a unique way to investigate the anatomical basis of brain reward function and motivated behavior. Secondly, eICSS has been extensively used to evaluate a drug’s rewarding or aversive effects [1,9,12,13]. Changes in the stimulation–response curve of eICSS are used to quantitatively measure drug-induced changes in BSR [4,6,14,15,16] (Figure 1). Thirdly, the ICSS procedure has also been used to evaluate the abuse potential of new psychoactive substances and the therapeutic potential of novel compounds for treating substance use disorders [1,10,15]. As many drugs of abuse enhance BSR in eICSS, the antagonism of novel compounds on addictive drug-enhanced BSR serves as a crucial indicator of therapeutic effects [16,17,18]. In addition to this two-dimensional functional curve shift model, a 3D reward-mountain model has been developed to measure reward-seeking behavior as a function of both the strength and cost of reward [19,20]. The 3D method appears to be more sensitive and informative than the 2D methods in measuring a drug’s effects on brain reward function, subjective effort costs, and/or the value of activities that interact with eICSS [19,21].

However, despite its broad implications, the lack of cell type specificity in eICSS has been a significant drawback in studying neural mechanisms underlying brain reward processes and addiction [1,22,23]. As the MFB contains both ascending and descending fiber projections between the brainstem and the forebrain and the VTA contains multiple phenotypes of neurons [24,25,26], electrical stimulating the MFB or VTA results in the release of various transmitters from diverse cell types in multiple brain regions.

The recent development of optogenetics has provided a promising solution to this challenge. By introducing opsins, light-sensitive proteins, researchers can make genetically defined populations of cells light-sensitive [27,28]. This revolutionary technique allows us to selectively manipulate (activate or inactivate) specific types of neurons or their projection terminals to determine their role in reward processes and addiction [23,29,30]. For instance, stimulation of the mesolimbic DA system with optogenetics has confirmed the crucial role of DA or DA-related circuits in reward and motivation [4,26,31,32,33]. In addition to DA, optogenetic stimulation of glutamate neurons or inactivation of GABA neurons in distinct brain regions has also been shown to produce optical ICSS (oICSS) [33,34,35,36,37], supporting positive reinforcement effects. However, it is worth noting that most studies utilizing optogenetics have employed a single stimulation frequency or intensity to determine which types of neurons, when activated, are able to produce rewarding effects. oICSS has not been used to evaluate the rewarding or aversive effects of a drug on oICSS until 2017 when we first introduced the stimulation frequency-rate response curve to oICSS, akin to the one used in eICSS (Figure 1). In this study, we found that drugs of abuse such as cocaine and cannabinoids differentially alter the brain reward function in transgenic VgluT2-Cre mice [14]. This innovative approach represents a significant step forward, as it allows us to quantitatively evaluate the effects of drugs of abuse on oICSS, mirroring the established methods in eICSS.

**Figure 1 ijms-25-03455-f001:**
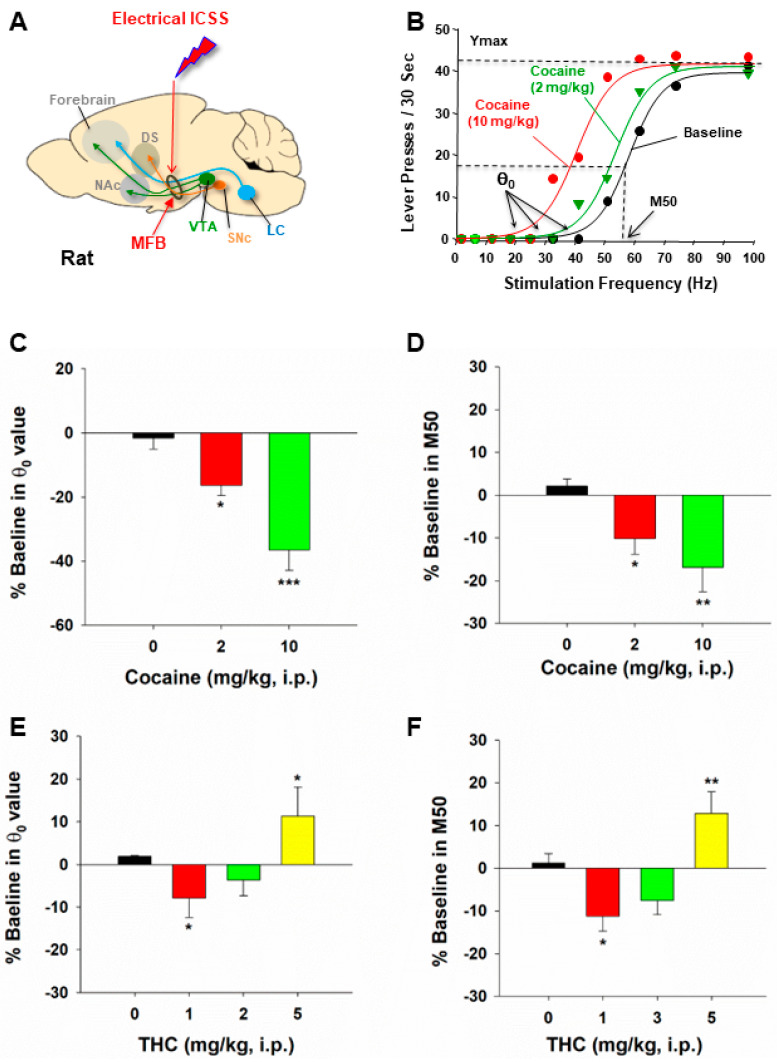
The effects of cocaine and THC on electrical brain-stimulation reward (BSR) in rats. (**A**) A diagram illustrates the medial forebrain bundle at the anterior–posterior level of the lateral hypothalamus and the location of a stimulation electrode for electrical BSR. (**B**) Representative stimulation–response curves, indicating that systemic administration of cocaine shifted the stimulation–response curve to the left and decreased the BSR stimulation threshold (θ_0_) and M50. (**C**,**D**) The effects of cocaine on the mean (±SEM) values of BSR stimulation threshold (θ_0_) (**C**) and M50 (**D**), indicating that cocaine dose-dependently shifted the frequency-rate response curve to the left and decreased the θ_0_ and M50 values. (**E**,**F**) The effects of THC on the mean (± SEM) values of θ_0_ (**E**) and M50 (**F**), indicating that THC produced biphasic effects—THC, at a low dose (1 mg/kg, i.p.) shifted the frequency-rate response curve to the left and decreased the θ_0_ and M50 values, while, at a high dose (5 mg/kg), THC shifted the curve to the right and increases θ_0_ and M50 values. * *p* < 0.05, ** *p* < 0.01, *** *p* < 0.001, compared to the vehicle control group; one-way ANOVA followed by post-hoc Student–Newman–Keuls tests for multiple group comparisons. The different colors represent different drug doses. Adapted from Xi et al., 2010 and Spiller et al., 2019 [15,38].

Compared to eICSS, oICSS offers significant advantages. Firstly, oICSS provides neuronal specificity, allowing researchers to target specific cell types or neural circuits with optogenetic techniques. This precision enables a precise examination of the underlying neural mechanisms involved in drug reward or aversion, enhancing our understanding of addiction and aversion neurobiology. Secondly, oICSS allows for temporal precision, enabling precise control over the timing and duration of neuronal activation, crucial for studying dynamic neural processes associated with drug-induced behaviors like reward learning and craving. Thirdly, oICSS facilitates circuit-level manipulation, enabling researchers to selectively modulate specific neural circuits implicated in drug reward and aversion pathways, elucidating the functional connectivity underlying drug-induced behaviors. In addition, oICSS allows for genetic targeting, enabling a more selective manipulation of neural activity, and potentially paving the way for gene-targeted therapeutic interventions.

In this mini-review article, we first review the principle of the frequency-rate response of oICSS for studying drug reward and aversion. Subsequently, we review recent progress in studying the effects of drugs of abuse and other test drugs on specific neurotransmitter-mediated oICSS. Lastly, we discuss the implications and constraints of oICSS in future studies.

## 2. Principles of oICSS for Studying Drugs of Abuse

The general experimental methods involve the introduction of opsins, light-sensitive proteins, into specific neurons or their projection terminals [27,28]. This genetic modification renders these populations of cells light-sensitive, allowing for precise control through optogenetic stimulation. The technique has been pivotal in studying cell type-specific neural mechanisms underlying brain reward function and motivation [31].

Traditionally, the rate-frequency curve-shift procedure has been a standard method in assessing drug effects in eICSS. In this procedure, the effects of drugs of abuse on eICSS behavior are evaluated through parameters such as θ_0_ (the minimally required stimulation frequency), Ymax (the maximal rate of lever response), and M50 (the stimulation frequency for half-maximal reward efficacy) (Figure 1B) [15,16,39]. Drugs of abuse, such as cocaine and amphetamine, cause a decrease in the stimulation threshold for electrical BSR and shift the stimulation–response curve leftward or upward immediately after acute administration. Similarly, systemic administration of GBR12935 (a selective DAT inhibitor) or SKF82958 (a DA D1R-like agonist) also produces a dose-dependent decrease in the BSR threshold and a leftward or upward shift of the eICSS curve [1,9]. These findings suggest that cocaine, DAT inhibitors, or D1R agonists each potentiate the rewarding effects of eICSS. In contrast, withdrawal from chronic cocaine or nicotine administration is associated with depression-like effects and deficits in brain reward function, as assessed by BSR threshold elevation or a rightward shift of eICSS [40,41]. Based on these findings, a well-accepted assumption is that if a test drug, such as cocaine, causes a decrease in θ_0_ and M50 values or a leftward or upward shift of the stimulation–response curve, it indicates enhanced BSR and a summation between BSR and drug reward [1,15,22] (Figure 1C,D). In contrast, if a drug, such as Δ^9^-THC, produces an increase in θ_0_ or M50 value or a rightward or downward shift in the ICSS curve, it is often interpreted as producing reward attenuation or aversive effects [1] (Figure 1E,F).

Similarly, the adoption of this approach in oICSS studies, with a focus on a shift of the stimulation-rate response of oICSS, provides a quantitative framework for evaluating drug-induced changes in neurotransmitter-dependent oICSS. The same assumption is used in oICSS. If a drug, such as cocaine, also causes a leftward or upward shift of the stimulation–response curve, it indicates enhanced BSR. Conversely, if a drug, such as Δ^9^-THC, produces a rightward or downward shift in the oICSS curve, it indicates the drug producing reward attenuation or aversive effects.

## 3. DA-Dependent oICSS and Its Implications in Studying Drug Reward versus Aversion

Extensive research has focused on the functional roles of VTA DA neurons in reward processes [4,6,42,43]. These neurons play distinct roles in both positive and negative reinforcement, resulting in preference and avoidance behaviors, respectively [23,31,42,44,45,46]. VTA DA neurons exhibit increased activity in response to both rewarding and aversive stimuli [46,47], suggesting physiological implications of these neurons in diverse and even conflicting environmental settings. Despite the complicated responses of DA neurons to seemingly conflicting cues, acute activation of these neurons often leads to positive reinforcement and behavioral preference [31,32].

In 2011, Witten et al. first reported that selective optogenetic activation of VTA DA neurons in TH-Cre mice supported oICSS [48]. Subsequent research over the following decade has confirmed previous findings regarding the role of DA neurons in the VTA and substantia nigra (SN) in driving positive oICSS and reinforcement in rats and mice [4,49,50,51,52] (Table 1, Figure 2). However, projection-specific optogenetic manipulations indicated that VTA dopaminergic projections to sub-regions of the NAc may play different roles in reinforcement—VTA neurons that primarily project to the NAc-core support positive oICSS, while a subpopulation of VTA neurons that project to the NAc-shell do not [53], revealing previously unexpected complexities in neural pathways of reinforcement.

**Table 1 ijms-25-03455-t001:** Neural substrates underlie oICSS when activated.

Animals	Brain Region	Targeted Neurons	Opsins	Stim. Frequency	Finding	References
Dopamine-dependent oICSS						
TH-Cre rats	VTA	DA	ChR2	20 Hz	Produces oICSS	[4,48,54]
TH-Cre mice	VTA, SNc	DA	ChR2	25 Hz	Produces oICSS	[52,55]
TH-Cre mice	VTA	DA	ChR2	20 Hz	Produces oICSS	[56]
TH-Cre	VTA	DA	ChR2	20 Hz	Produces oICSS	[57,58,59,60,61]
DAT-Cre mice	VTA-NAc	DA terminals	ChR2	30 Hz	Produces oICSS	[62]
DAT-Cre	SNc	DA	ChR2	50 Hz	Produces oICSS	[63]
DAT-Cre mice	VTA	DA	ChR2	1, 5, 10, 25, 50, 100 Hz	Produces oICSS	[17,18,49,50,51]
DAT-Cre, Crhr1-, Cck-, mice	VTA	DA	ChR2	20 Hz	Produces oICSS	[53]
DAT-Cre mice	VTA	DA	ChR2	1, 5, 10, 20, 25, 50 Hz	Produces oICSS	[64]
DAT-Cre mice	VTA	DA	ChR2	1, 5, 10, 20, 25, 50, 65 Hz	Produces oICSS	[65]
DAT-Cre mice	VTA	DA	ChR2	20 Hz	Produces oICSS	[66]
DAT-Cre mice	NAc, PFC	DA terminals	ChR2	20 Hz	Produces oICSS	[66]
DAT-Cre	VTA	DA	ChR2	40 Hz	Produces oICSS	[67]
D1-Cre	Dentate gyrus	DA			Produces oICSS	
Glutamate-dependent oICSS						
C57 WT mice	BLA-VTA	Glutamate	ChR2	20 Hz	Produces oICSS	[68]
C57 WT mice	vHipp-NAc	Glutamate	ChR2	20 Hz	Produces oICSS	[69]
VgluT2-Cre mice	VTA	Glutamate	ChR2	20 Hz	Produces oICSS	[35]
Thy1-ChR2-EYFP *mice*	NAc	Glutamate terminals	ChR2	20 Hz	Produces oICSS	[70]
VgluT2-Cre mice	Pedunculopontine	Glutamate	ChR2	10, 20, 30, 40 Hz	Produces oICSS	[71]
VgluT2-Cre mice	VTA	Glutamate	ChR2	1, 5, 10, 25, 50, 100 Hz	Produces oICSS	[14]
VgluT2-Cre mice	DMS,	Glutamate terminals	ChR2	20 Hz	Produces oICSS	[72]
VgluT2-Cre mice	NAc	Glutamate terminals	ChR2	40 Hz	Produces oICSS	[73]
VgluT2-Cre mice	VTA	Glutamate	ChR2	10, 20, 30, 40 Hz	Produces oICSS	[36]
VgluT2-Cre mice	VP, NAc, LHb	Glutamate terminals	ChR2	10, 20, 30, 40 Hz	Produces oICSS	[36]
VgluT2-Cre mice	RN	Glutamate	ChR2	20 Hz	Produces oICSS	[17]
VgluT2-Cre mice	VTA	Glutamate terminals	ChR2	20 Hz	Produces oICSS	[17]
VgluT2-Cre mice	Parabrachio-SNc	Glutamate terminals	ChR2	20 Hz	Produces oICSS	[74]
GABA-dependent oICSS						
C57 WT	NAc	GABA	ChR2	20 Hz	Produces oICSS	[69]
Vgat-Cre	SNr,	GABA	Halo	Constant, 20 s	Produces oICSS	[51]
Vgat-Cre	SNr	GABA	Arch3	Constant, 3 s	Produces oICSS	[75]
D1-Cre mice	DS	D1-MSNs	ChR2	Constant, 1 s	Produces oICSS	[76]
D1-Cre mice	DS	D1-MSNs	ChR2	40 Hz	Produces oICSS	[75]
D1-Cre mice	DS	D1-MSNs	ChR2	5 Hz	Produces oICSS	[77]
D1-Cre mice	NAc	D1-MSNs	ChR2	25 Hz	Produces oICSS	[78]
Other substance-dependent oICSS						
ePet-Cre mice	DRN	5-HT	ChR2	5, 20 Hz	Produces oICSS	[79]
ePet-Cre mice	DRN	5-HT	ChR2	20 Hz	Produces oICSS	[56]
SERT-Cre mice	DRN, 5-HT neurons	5-HT	ChR2	40 Hz	Produces oICSS	[75]
Tac2-Cre mice	dMHb	Neurokinin-expressing	ChR2	20 Hz	Produces oICSS	[80]
D1-Cre	LC-DG	D1-expressing	ChR2	20 Hz	Produces oICSS	[81]

Notes: VTA, ventral tegmental area; NAc, nucleus accumbens; DS, dorsal striatum; SNc, substantia nigra pars compacta; SNr, substantia nigra pars reticulata; PFC, prefrontal cortex; BLA, basolateral amygdala; DRN, dorsal raphe nucleus; LC, locus coeruleus; DG, dentate gyrus; dMHb, dorsal medial habenula; D1-MSNs, D1 receptor-expressing medium-spiny neurons; 5-HT, serotonin; DAT-Cre, Cre recombinase expressed in dopamine transporter (DAT)-expressing neurons; TH-Cre, Cre recombinase expressed in tyrosine hydroxylase (TH)-expressing neurons; ePet-Cre, Cre recombinase expressed in serotoninergic neurons; SERT-Cre, Cre recombinase expressed in serotonin transporter (SERT)-expressing neurons; VgluT2-Cre, Cre recombinase expressed in type 2 vesicular glutamate transporter (VgluT2)-expressing neurons; Vgat-Cre, Cre recombinase expressed in vesicular GABA transporter (GAT)-expressing neurons.

**Figure 2 ijms-25-03455-f002:**
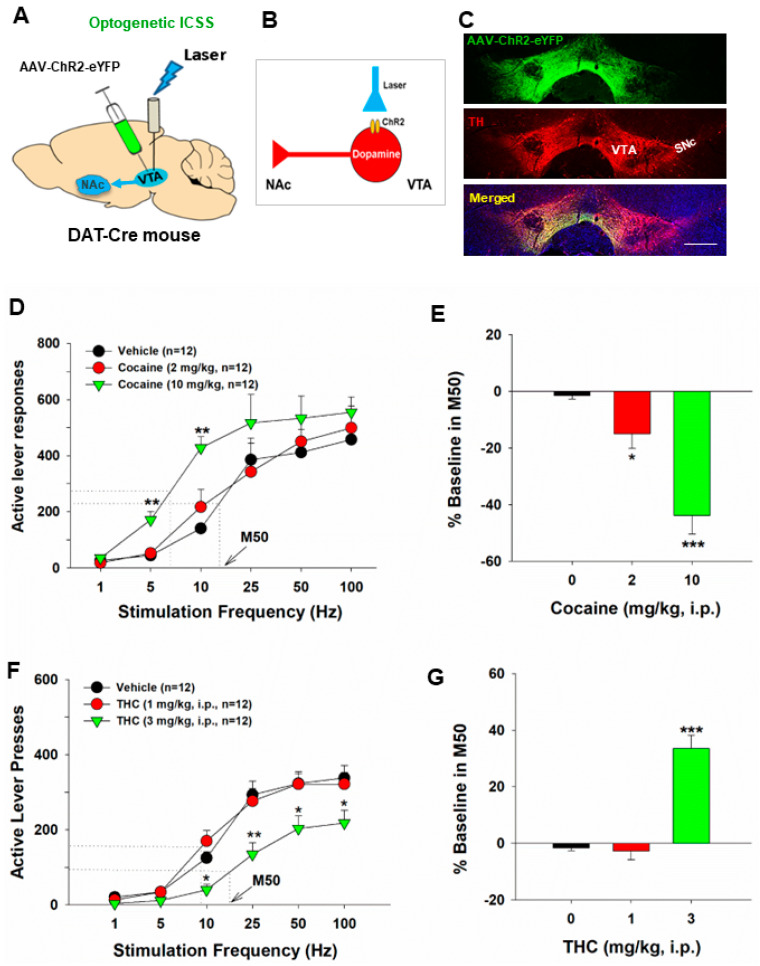
Optical intracranial self-stimulation (oICSS) experiment in DAT-Cre mice. (**A**) Schematic diagrams illustrating that AAV-ChR2-eYFP vectors were microinjected into the lateral VTA and optical fibers (i.e., optrodes) were implanted in the same brain region. (**B**) A diagram illustrates how AAV-ChR2 is expressed on VTA DA neurons, which can be activated by a 473 nm laser. (**C**) Representative images of AAV-ChR2-eYFP and TH expression in the VTA. The scale bar indicates 200 μM. (**D**) the stimulation-rate response curves, indicating that optogenetic activation of VTA DA neurons induced robust oICSS behavior (lever presses) in DAT-Cre mice in a stimulation frequency–dependent manner. Systemic administration of cocaine shifted the frequency–rate response curve to the left and decreased M50 values. (**E**) Cocaine-induced % changes in M50 over pre-cocaine baseline. (**F**) Effects of THC on DA-dependent oICSS. Systemic administration of THC dose-dependently shifted the curve to the right and increased M50 values. (**G**) THC–induced % changes in M50 over pre-THC baseline. * *p* < 0.05, ** *p* < 0.01, *** *p* < 0.001, compared with the vehicle control group. The different colors represent the different treatment or drug dose groups. Adapted from Hempel et al., 2023 and Jordan et al., 2020 [82,83].

As stated above, the frequency-rate eICSS procedures have been used as a tool for testing drug abuse potential for decades. Several previous review articles have summarized the major findings of the effects of drugs of abuse on eICSS [1,2,6,11,22]. Most previous studies observed the effects of acute drug administration on eICSS [1,9]. For example, acute administration of cocaine or methamphetamine produced a dose-dependent leftward or upward shift in the frequency-rate curve [1]. In contrast, opioids produced mixed effects—highly addictive opioids, such as morphine and fentanyl, weakly facilitated eICSS at low doses; but, at higher doses, they produced initial eICSS depression followed later by eICSS facilitation [1,9]. Cannabinoids with CB1 and CB2 receptor agonist profiles produced little, biphasic, or depression of eICSS in rats. An early study reported that Δ^9^-THC facilitated eICSS in Lewis rats [84]. A later study from the same group found that Δ^9^-THC facilitated eICSS in Lewis and Sprague–Dawley rats but not in Fischer 344 rats [85]. In contrast, other studies found that Δ^9^-THC produced a dose-dependent biphasic effect—low doses facilitated, while high doses depressed eICSS in Sprague–Dawley or Long-Evans rats [15,86] or produced monophasic dose-dependent depression of eICSS in Sprague–Dawley rats [13,87]. Consistent with the latter finding, other cannabinoid agonists such as nabilone, levonantradol, CP55940, WIN55212-2, and HU210 produced only dose-dependent depression of ICSS in rats of various strains [12,87,88].

Utilizing the same frequency-rate response of oICSS, we investigated the impact of drugs of abuse on DA-dependent oICSS, yielding a series of novel findings. Acute administration of cocaine resulted in dose-dependent oICSS facilitation, evidenced by a leftward or upward shift and a reduction in M50 [18,49,50,89] (Figure 2). Similarly, opioids such as oxycodone demonstrated dose-dependent biphasic effects—low doses facilitated, while high doses inhibited oICSS in DAT-Cre mice [90]. These findings align with observations in eICSS in rats. Notably, the BSR-enhancing effects of cocaine were more potent at low electrical or optical stimulation frequencies, suggesting that cocaine-enhanced extracellular DA via blockade of the DA transporter (DAT) may exhibit additive or synergistic effects with DA neuron activation produced by low-frequency stimulation.

We also employed oICSS as a novel behavioral tool to assess the abuse potential of novel DAT inhibitors such as JJC8-088 and JJC8-091. JJC8-088 induced a cocaine-like upward or leftward shift in oICSS in DAT-Cre mice, implying potential cocaine-like abuse [89]. In contrast, JJC8-091 prompted a contrary downward shift in oICSS, suggesting potential therapeutic anti-cocaine properties [89]. Indeed, results from a series of behavioral, neurochemical, and electrophysiological experiments substantiate the findings and conclusions observed in oICSS [89].

Furthermore, we utilized oICSS to investigate the potential abuse or aversive effects of novel DA D3 receptor ligands on DA-dependent oICSS behavior. Novel D3 receptor ligands (±)VK4-40 (a D3 receptor antagonist), R-VK4-40 (also a D3 receptor antagonist), and S-VK4-40 (a D3 receptor partial agonist) induced mild depression in oICSS, while their pretreatment functionally counteracted cocaine- or oxycodone-enhanced oICSS [90,91,92]. These findings not only suggest an essential role of the D3 receptor in mediating DA-mediated oICSS through the blockade of D3 receptors or by competing with excess DA binding to D3 receptors caused by optical stimulation but also provide additional evidence supporting the utility of these D3 receptor ligands for treating substance use disorders, with a potential low abuse risk themselves.

Additionally, we extensively employed this behavioral model to investigate the functional role of cannabis or cannabinoids on DA-dependent behavior. Our findings revealed that systemic administration of Δ^9^-THC, WIN55,212-2, but not cannabidiol, dose-dependently decreased oICSS and shifted oICSS curves downward. Similarly, cannabinoid ligands that selectively activated CB1 (by ACEA), CB2 (by JWH133), or PPARγ (by pioglitazone) also induced dose-dependent reductions in oICSS [18,50,82]. Pretreatment with antagonists of CB1 (AM251, PIMSR), CB2 (AM630), or PPARα/γ (GW6471, GW9662) receptors dose-dependently blocked Δ^9^-THC-induced reduction in oICSS [18,82]. However, when examining various new synthetic cannabinoids in this behavioral model, we found that XLR-11 produced a cocaine-like enhancement, AM-2201 produced a Δ^9^-THC-like reduction, and 5F-AMB had no effect on oICSS [50]. Together, these findings from oICSS suggest that most cannabinoids are not rewarding or reward-enhancing but rather reward-attenuating or aversive in mice, and multiple cannabinoid receptor mechanisms underlie cannabinoid action in DA-dependent behavior. These findings not only confirm some previous findings with eICSS but also expand our understanding of the role of DA in cannabinoid action.

This newly established behavioral model has also been instrumental in cannabis-based medication development for treating substance use disorders (Table 2). In recent years, the cannabinoid CB2 receptor has emerged as a new target in medication development for the treatment of substance use disorders, as this receptor has been identified on midbrain DA neurons and implicated in drug reward and addiction [93,94]. Systemic administration of beta-caryophyllene (BCP), a plant-derived product with a CB2 receptor agonist profile and also an FDA-approved food additive, mildly depressed oICSS in DAT-Cre mice [95,96,97]. Pretreatment with BCP dose-dependently inhibited oICSS-enhancing effects produced by cocaine, methamphetamine, and nicotine [95,96,97]. Although the neural mechanisms underlying BCP action in oICSS are not fully understood, the simplest explanation is that activation of CB2 receptors on DA neurons inhibits DA neuron activity [98,99], subsequently counteracting the DA-enhancing effects produced by optogenetic stimulation of DA neurons or drugs of abuse.

## 4. Glutamate-Dependent oICSS and Its Application in Studying Drug Reward and Addiction

The VTA is well-known for regulating reward consumption, learning, memory, and addiction [26,73,100,101,102]. In addition to DA neurons, the VTA contains other types of neurons, including glutamate neurons and GABA neurons [26]. Unlike the well-studied functions of DA neurons, the role of VTA glutamate neurons is understudied. However, emerging studies have begun to reveal the importance of glutamate in regulating reward processes and addiction. In the brain, glutamate is synthesized from glutamine by glutaminase and then packaged into vesicles by vesicular glutamate transporters (VgluT) for its synaptic release [103]. Glutamate neurons in the VTA mainly express VgluT2 but not VgluT1 or VgluT3 [104,105]. VgluT2-expressing glutamate neurons are mostly located in the anterior and middle line of the VTA [49,104] and project to the NAc, ventral pallidum (VP), PFC, dorsal hippocampus (DH), and lateral habenula (LHb) [14,36,73,105,106] (Figure 3).

Optical stimulation of VTA glutamate neurons is rewarding, as assessed by the increased firing of VTA DA neurons, supporting oICSS, and producing conditioned place preference and appetitive instrumental conditioning [14,35,36]. The rewarding effects of VTA glutamate neurons are suggested to be mediated via a local excitatory synapse connection between VTA glutamate and DA neurons [14,35]. This is further supported by our finding that pretreatment with DA D1 or D2 receptor antagonists attenuates oICSS maintained by optical stimulation of VTA glutamate neurons in VgluT2-Cre mice [14]. Additionally, optical activation of VTA glutamate neurons could also support oICSS in the absence of DA release [73], suggesting a DA-independent mechanism underlying glutamate-mediated reward. In contrast to reward, evidence also shows that optical stimulation of VTA glutamate neurons induces aversive escape behaviors [102], and optogenetic stimulation of VTA glutamatergic terminals in the NAc induces aversion [30,107].

The LHb is a brain region known for its function in conditioning aversion and reward [108,109]. Optogenetic activation of VTA glutamatergic terminals in the LHb elicits aversion and produces aversive conditioning [110]. These findings suggest that in addition to local glutamate projections within the VTA, VTA glutamate neurons also project to other brain regions and activation of distinct glutamate pathways may produce rewarding or aversive effects.

We have recently utilized glutamate-dependent oICSS to assess the rewarding versus aversive effects of drugs of abuse in VgluT2-Cre mice. Our findings indicate that systemic administration of cocaine caused a significant leftward shift of the rate-frequency curve of oICSS, suggesting a reward-enhancing effect [14]. This observation aligns with the results observed in DAT-Cre mice [18,51,89]. As DA receptor antagonists significantly attenuated oICSS and shifted the rate-frequency curve to the right [14], it suggests that the oICSS produced by the activation of VTA glutamate neurons is at least partially mediated by the activation of VTA DA neurons. This DA-dependent mechanism may also explain how acute cocaine produces an enhancement in glutamate-mediated oICSS, as cocaine is an indirect DA enhancer through pharmacological blockade of DAT in the NAc.

We also employed this glutamate-dependent oICSS behavioral model to investigate the rewarding versus aversive effects of cannabinoids. Cannabis can elicit both rewarding and aversive responses in both humans and experimental animals. Cannabis reward is believed to be mediated by the activation of cannabinoid CB1 receptors on GABAergic neurons, leading to the disinhibition of VTA DA neurons [111]. However, there is a lack of direct behavioral evidence supporting this GABAergic hypothesis. To address this, we recently used RNAscope in situ hybridization assays to examine the cellular distribution of CB1 receptors in the brain. Our findings revealed that CB1 receptors are not only expressed on VTA GABAergic neurons but also on VTA VgluT2-positive glutamatergic neurons [14]. We then used Cre-Loxp transgenic technology to selectively delete CB1 receptors from glutamatergic neurons or GABAergic neurons. The results showed that systemic administration of Δ^9^-THC produced a dose-dependent conditioned place aversion and a reduction in glutamate-mediated oICSS in VgluT2-cre control mice, but not in glutamatergic CB1-KO mice [14]. These findings, for the first time, suggest that the activation of CB1 receptors expressed in VgluT2-positive glutamate neurons contributes to the aversive effects of cannabis or cannabinoids.

It is well-known that opioids are rewarding and produce analgesic effects. Opioid reward has been thought to be mediated through the inhibition of GABA transmission and subsequent disinhibition of DA neurons [112,113]. Recent studies indicate that mu-opioid receptors are not only expressed in VTA GABA neurons but also in VTA glutamate neurons [51,114]. Optogenetic activation of VTA glutamate neurons resulted in excitatory currents recorded from VTA DA neurons that were reduced by presynaptic activation of the mu-opioid receptor ex vivo [114]. In addition, opioid administration also directly inhibits glutamatergic transmission [115]. These findings suggest an important role of glutamate neurons in opioid effects. Opioids, such as morphine or oxycodone, have been shown to strengthen glutamatergic inputs to VTA DA neurons [114,116,117]. Furthermore, optically activating the VTA glutamate terminals in the dorsal hippocampus promotes opioid preference [107]. Together, growing evidence suggests that VTA glutamate neurons are also involved in opioid effects. oICSS may be used to study the functional role of glutamate neurons in opioid reward.

## 5. GABA-Dependent oICSS

VTA GABA neurons also play a crucial role in modulating reward consumption, depression, stress, and sleep by forming local synapses onto DA neurons [51,118,119] or by sending GABAergic projections to other brain regions such as the NAc, PFC, central amygdala, and dorsal raphe nucleus [119,120,121,122]. While anatomical and electrophysiological data reveal that VTA GABA neurons form local synapses onto other VTA neurons [123,124], the inhibitory input from local GABA neurons is much weaker than long-range GABAergic inhibitory inputs from the RMTg and NAc [122,124]. Evidence has shown that disrupting local GABA release within the VTA causes major malfunctions in stress and anxiety modulation through a DA-dependent mechanism [118,119,125,126]. Optogenetic stimulation of VTA GABA neurons also induces conditioned place aversion [127] and disrupts cocaine and sucrose reward consummation [51,118]. As optogenetic stimulation of VTA GABA neurons directly suppressed the activity and excitability of neighboring DA neurons, as well as the release of DA in the NAc [118,127], it is suggested that the dynamic interplay between VTA DA and GABA neurons can control the initiation and termination of reward-related behaviors. However, direct optical activation of VTA GABA failed to alter heroin self-administration in Vgat-Cre mice [51], suggesting that VTA GABA interneurons may play a limited role in opioid reward.

In addition to VTA, the striatum consists of multiple types of neurons, including a large population (~95%) of GABAergic medium spiny neurons (MSNs) and a smaller population of interneurons [128]. The MSNs are classified as D1-MSNs and D2-MSNs based on the expression of D1 or D2 receptors [129]. Interestingly, optogenetic stimulation of D1-MSNs within the dorsal striatum [75,76,77], NAc [78], or olfactory tubercle [130] is positively reinforcing, as assessed by real-time place preference or oICSS, while optogenetic stimulation of D2-MSNs within the dorsal striatum causes conditioned place aversion [76]. The reinforcing effects of D1-MSNs have been shown to be mediated through GABAergic projections from the dorsal striatum to the substantia nigra pars reticulata (SNr) to the ventromedial motor thalamus (VMT) [75]. We have recently reported that optogenetic stimulation of SNr GABA neurons is not rewarding in real-time place preference [47] (Figure 4). In contrast, optogenetic inhibition of SNr GABA neurons produced real-time place preference and oICSS [47] (Figure 4). These findings suggest that the rewarding effects of optical stimulation striatal D1-MSNs could be mediated by GABA-mediated inhibition of SNr GABA neurons that subsequently disinhibits glutamatergic neurons in the VMT [75]. Taken together, these in vivo optogenetic experiments demonstrated that optogenetic activation of striatal GABA neurons or optogenetic inhibition of SNr GABA neurons supports positive oICSS (Table 1). So far, the frequency-rate response curve has not been used in this behavioral model to study the effects of drugs of abuse or other drugs on GABA-mediated oICSS.

## 6. Advantages and Limitations of oICSS

Intravenous (i.v.) drug self-administration, conditioned place preference (CPP) or aversion (CPA), and intracranial self-stimulation (ICSS) are the most used behavioral procedures to assess the rewarding effects of drugs of abuse [1]. I.V. self-administration measures drug reinforcement through lever pressing for drug infusion. CPP/CPA assesses drug-associated context preference or aversion. ICSS measures reward by delivering electrical or optical stimulation to specific brain regions (Table 3). While i.v. self-administration mimics human drug-taking, CPP evaluates conditioned aspects of drug reward, and ICSS targets specific neural circuits. All three methods contribute to understanding drug reward and addiction, with i.v. self-administration being most translational, CPP revealing conditioned aspects, and ICSS providing neurobiological insights (Table 3).

In comparison to classical eICSS, the oICSS procedure presents several advantages. Firstly, oICSS behavior is cell-type specific. oICSS allows precise targeting of specific neural populations such as midbrain DA neurons, glutamate neurons, or GABA neurons, providing insights into the role of distinct cell types and neural circuits in reward processing [27,28]. For example, the findings from the oICSS in the above-mentioned Cre-mouse lines not only confirm previous findings regarding the role of DA in reward and motivation but also reveal unexpected new findings indicating that multiple neurotransmitters and circuits underlie brain reward function. Secondly, oICSS has a high temporal resolution, which enables precise control over the timing of neural activation, facilitating the study of temporal dynamics in reward-related behaviors. Thirdly, oICSS can be combined with other behavioral paradigms such as i.v. self-administration and RTPP, providing flexibility in experimental design, and allowing for the investigation of different aspects of reward processing. Fourthly, oICSS can be integrated with imaging techniques such as calcium imaging or fiber photometry, enabling real-time monitoring of neural activity during reward-related behaviors. And lastly, oICSS is relatively safer than eICSS for in vivo experiments. Little evidence indicates that photostimulation (10–20 mW) of ChR2-expressing neurons leads to significant tissue damage or cell death [23]. In addition, oICSS appears to be more robust and stable over time than classical eICSS based on our over 10 years of working experience in oICSS. Mice quickly learn to lever press for oICSS, and once they have acquired the behavior, responding may last longer time (up to 5 months), whereas eICSS behavior in rats usually lasts 1–2 months. Thus, it enables the testing of multiple drugs in the same subjects (with appropriate washout periods), with the added benefit of reducing animal numbers. Therefore, oICSS could be especially suitable for screening a large number of compounds for abuse potential.

Given the multifaceted mechanisms occurring at molecular, cellular, and circuit levels with drugs of abuse, oICSS study can provide more insight into these processes. At the molecular level, oICSS can modulate neurotransmitter release, receptor activation, and intracellular signaling pathways within targeted brain regions. At the cellular level, oICSS can alter the excitability and firing patterns of specific neuronal populations, leading to changes in synaptic strength, neuronal plasticity, and network activity. This may involve the activation or inhibition of distinct cell types within neural circuits associated with drug reward and aversion, influencing their overall functional dynamics. At the circuit level, oICSS can selectively modulate the activity of interconnected brain regions involved in the processing of reward-related stimuli and the regulation of motivated behavior. By targeting specific neural circuits implicated in drug reward and aversion pathways, oICSS allows researchers to dissect the functional connectivity and causal relationships underlying these complex behaviors. 

The limitations of oICSS as a new behavioral model to evaluate drug-rewarding versus aversive effects include the use of transgenic Cre-expressing mouse or rat lines, AAV vector microinjections, and transgenic opsin expression. The availability of transgenic animals may limit the use of oICSS, and the differences in opsin expression levels may impact the basal levels of oICSS behavior between studies, making original data comparisons between studies difficult [131]. In addition, stimulation threshold (θ_0_) and M50 values that are routinely used in eICSS have not been used in oICSS to evaluate drug effects. In eICSS, 16 different electrical pulse frequencies ranging from 141 to 25 Hz are used to generate a stimulation–response curve, allowing us to accurately calculate θ_0_ and M50 using best-fit mathematical algorithms as reported previously [15,16]. However, in oICSS, there are only 6 different laser pulse frequencies ranging from 1 to 100 Hz for establishing a stimulation–response curve. Thus, more efforts are needed to optimize the oICSS procedure. Furthermore, optogenetic stimulation of nerve terminals may generate back-propagating action potentials, which can secondarily activate additional projections of a particular cell [132]. Thus, neurotransmitter release in other projection regions of a cell may complicate the data explanations.

The other limitations include limited spatial resolution, the requirement for technical expertise, and costly setup and maintenance. While offering cell-type specificity, optogenetic activation might not perfectly replicate the natural firing patterns of neurons, potentially introducing unintended consequences. It is also important to note that oICSS may not fully replicate the effects of drug abuse observed in preclinical and clinical settings. The intense stimulation or inhibition induced by oICSS may disrupt brain circuitry and impact behavior significantly, potentially leading to non-physiological responses or alterations in neurotransmitter dynamics that do not accurately reflect the complexities of drug-induced behaviors. Furthermore, the interpretation of oICSS findings in the context of drug abuse or aversion requires careful consideration of the limitations and potential confounding factors associated with this technique. These may include long-lasting compensatory responses that may arise in response to chronic or repeated stimulation. Despite these limitations, oICSS remains a valuable tool for researchers due to the increased precision and control it offers. As the technology continues to develop, these limitations might be addressed in the future.

We also note that although ICSS procedures have been commonly used to examine the abuse potential of drugs, these procedures have neither been listed as standard drug screening tests in the field of drug abuse and addiction [133,134] nor listed in the FDA’s Guidance for Industry for assessing the abuse potential of drugs for regulatory purposes [135]. We live in an age of proliferating drug development that requires an expanding capability for abuse potential testing. This highlights the importance of improving predictive validity for oICSS as a viable tool in screening the abuse potential of drugs in future studies.

## 7. Potential Implications of oICSS in Human Studies

While optogenetic ICSS offers exciting possibilities for studying reward and addiction in rodents, its application in human studies presents significant ethical and practical challenges.

One of the most significant implications of oICSS in human studies lies in the development of targeted therapies for addiction. Understanding the precise neural circuits underlying addictive behaviors could facilitate the development of more effective treatments for drug addiction and behavioral addictions, ultimately improving outcomes for individuals struggling with these disorders. By targeting specific brain regions implicated in addiction, oICSS may pave the way for personalized therapies tailored to individual patients’ unique neural circuitry, offering hope for more effective and personalized treatment approaches.

Furthermore, oICSS has the potential to shed light on the neural basis of other mental illnesses characterized by dysfunctions in reward processing, such as depression or schizophrenia. By unraveling the complex interplay of neural circuits underlying these disorders, oICSS could provide insights into novel treatment targets and therapeutic approaches, advancing our understanding and management of these debilitating conditions.

Additionally, oICSS techniques allow for sophisticated manipulation and interrogation of neural circuits in animal models. Validation of these findings in human studies is crucial for enhancing the translational relevance of preclinical research and improving our understanding of the neurobiological basis of behavior and disease. By bridging the gap between animal models and human studies, oICSS holds the potential to accelerate the development of novel therapeutics for addiction and other neuropsychiatric disorders.

In summary, the evolution of ICSS from its inception in the 1950s to the integration of optogenetics in recent years represents a continuum of advancements that have significantly enriched our understanding of the neural mechanisms underlying reward processes and addiction. The oICSS procedure has not only served as a tool to investigate the anatomical basis of brain reward function and motivated behavior but has also been crucial in evaluating the effects of drugs of abuse and new psychoactive substances. The recent integration of optogenetics has further refined our ability to selectively manipulate specific neural circuits, offering a more precise understanding of the neural basis of reward and motivation. As research in this field continues to evolve, the combination of traditional ICSS methods and cutting-edge techniques such as optogenetics holds immense promise for unraveling the complexities of the brain’s reward system and developing targeted interventions for substance use disorders.

## Figures and Tables

**Figure 3 ijms-25-03455-f003:**
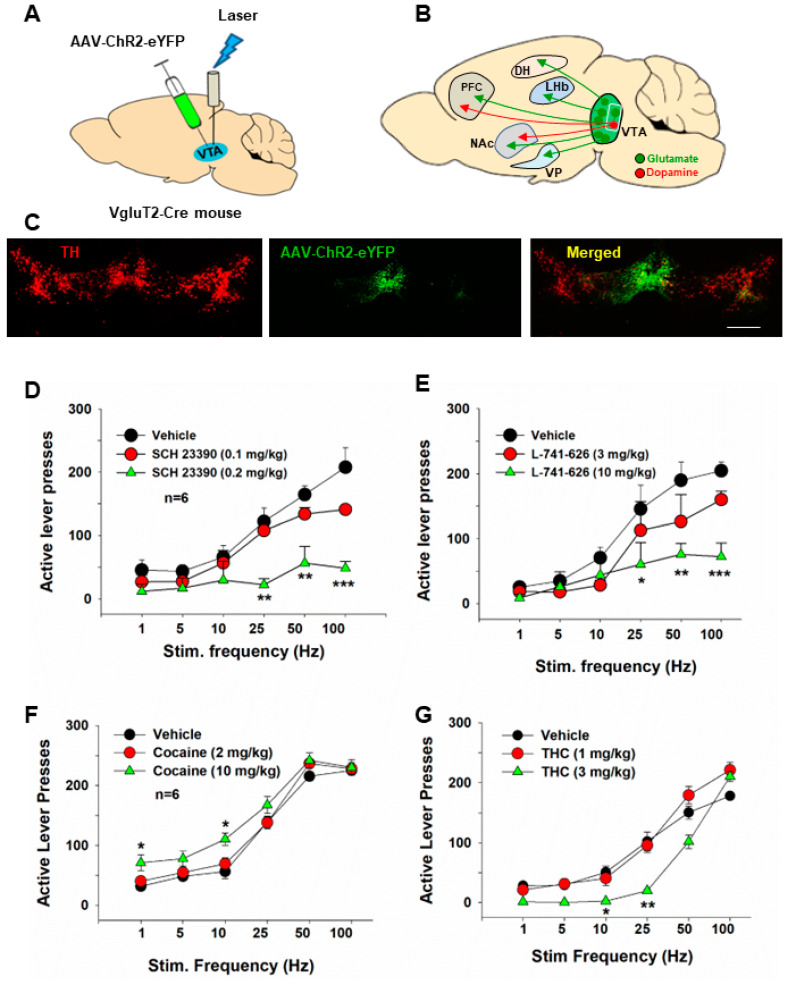
Optical intracranial self-stimulation (oICSS) experiment in VgluT2-Cre mice. (**A**) Schematic diagrams illustrating the target brain region (VTA) of the AAV-ChR2-GFP microinjection and intracranial optical fiber implantation. (**B**) Schematic diagram showing VTA glutamate projections. Within the VTA, some glutamate neurons locally synapse onto DA neurons. (**C**) Representative images of AAV-ChR2-EGFP expression in the medial VTA. The scale bar indicates 200 μM. (**D**,**E**) The stimulation frequency-rate response curve, indicating that optical stimulation of VTA glutamate neurons produced robust oICSS in VgluT2-Cre mice in a stimulation frequency-dependent manner. Systemic administration of SCH23390, a selective D1 receptor antagonist significantly inhibited the oICSS (**D**), while L-741,626, a selective D2 receptor antagonist, also dose-dependently inhibited the oICSS (**E**). (**F**,**G**) Systemic administration of cocaine dose-dependently shifted the rate-frequency function curve leftward and upward in VgluT2-Cre mice (**F**), while THC produced the opposite effect, producing a dose-dependent rightward shift (**G**). * *p* < 0.05, ** *p* < 0.01, *** *p* < 0.001, compared with the vehicle control group. Adapted from Han et al., 2017 [14].

**Figure 4 ijms-25-03455-f004:**
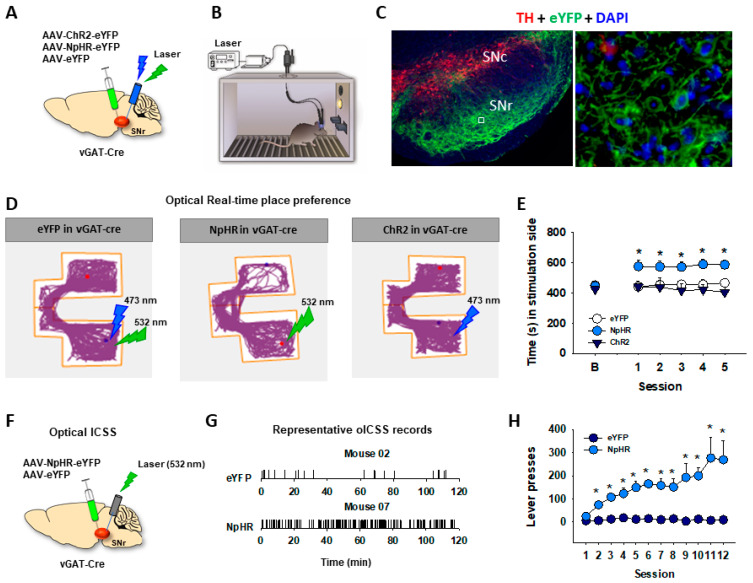
Optical inhibition of GABA neurons in the substantia nigra reticulata is rewarding as assessed by real-time place preference (RTPP) and optical ICSS (oICSS). (**A**,**B**) The general experimental procedures. (**C**) Representative images, illustrating AAV-NpHR-eYFP (green) expression in SNr GABA neurons, not in TH^+^ (red) DA neurons. (**D**) Representative locomotor tracing records, indicating that optical inhibition of SNr GABA neurons was rewarding in vGAT-Cre mice, whereas optical activation of SNr GABA neurons in vGAT-Cre mice was neither rewarding nor aversive. vGAT-Cre mice with NpHR-eYFP expression in SNr GABA neurons spent more time on the green laser (532 nm)-paired compartment (**middle** panel), but no change in place preference was observed in vGAT-Cre mice transfected with ChR2-eYFP in SNr GABA neurons (**right** panel). Laser stimulation (either 473 nm or 532 nm) had no effect in vGAT-Cre mice transfected the control AAV-eYFP (**left** panel). (**E**) Optical RTPP across 5 consecutive test sessions, illustrating that only vGAT-Cre mice with NpHR expression in SNr GABA neurons showed significant laser-paired place preference. (**F**) The general experimental procedures of oICSS. (**G**) Representative oICSS records, illustrating that vGAT-Cre mice with intra-SNr NpHR microinjections exhibited robust oICSS, whereas in mice with intra-SNr AAV-eYFP control virus microinjections did not. (**H**) The time courses of oICSS during 12 d of oICSS training, indicating that optical inhibition of SNr GABA neurons is rewarding. A small white box in **C** shows the area where the right high-magnification image was taken. * *p* < 0.05 as compared to the baseline (**D**) or eYFP control virus group (**E**). Adapted from Galaj et al. [51].

**Table 2 ijms-25-03455-t002:** Effects of drugs of abuse, dopaminergic ligands, and cannabinoids on oICSS.

Tested Drugs	Mice	oICSS	Major Findings	References
Drugs of abuse				
Cocaine (5, 10, 15, 20 mg/kg)	DAT-Cre	20 Hz	↓ oICSS by 20 Hz laser	[58]
Heroin (1, 2, 4, 8, 16, 32 mg/kg)	DAT-Cre	20 Hz	↓ oICSS by 20 Hz laser	[60]
Cocaine (2, 10 mg/kg)	DAT-Cre	F-R curve	↑ oICSS, Shift the F-R curve to the left	[83,91]
Oxycodone (0.3, 1, 3 mg/kg)	DAT-Cre	F-R curve	↑ oICSS at low doses, ↓ oICSS at high doses, Shift the F-R curve upward or downward	[90]
DAT inhibitors				
JJC8-088 (DAT inhibitor)	DAT-Cre	F-R curve	↑ oICSS, Shift the F-R curve upward	[89]
JJC8-091 (Atypical DAT inhibitor)	DAT-Cre	F-R curve	↓ oICSS, Shift the F-R curve downward	[89]
Dopamine D3 receptor ligands				
(±)-VK4-40 (D3 antagonist)	DAT-Cre	F-R curve	↓ oICSS, Shift the F-R curve downward	[91]
R-VK4-40 (D3 antagonist)	DAT-Cre	F-R curve	↓ oICSS, Shift the F-R curve downward	[90]
S-VK4-40 (D3 partial agonist)	DAT-Cre	F-R curve	↓ oICSS, Shift the F-R curve downward	[92]
Cannabinoid receptor agonists				
THC	DAT-Cre	F-R curve	↓ oICSS, Shift the F-R curve downward	[49]
THC	VgluT2-Cre	F-R curve	↓ oICSS, Shift the F-R curve downward	[50]
WIN55,212-2	DAT-Cre	F-R curve	↓ oICSS, Shift the F-R curve downward	[50]
AM-2201	DAT-Cre	F-R curve	↓ oICSS, Shift the F-R curve downward	[50]
CBD	DAT-Cre	F-R curve	No effect on oICSS	[50]
ACEA (CB1 agonist)	DAT-Cre	F-R curve	↓ oICSS, Shift the F-R curve downward	[50]
JWH133 (CB2 agonist)	DAT-Cre	F-R curve	↓ oICSS, Shift the F-R curve downward	[49]
Xie2-64 (CB2 inverse agonist)	DAT-Cre	F-R curve	↓ oICSS, Shift the F-R curve downward	[83]
BCP (CB2 agonist)	DAT-Cre	F-R curve	↓ oICSS, Shift the F-R curve downward	[95,96,97]
GW7647 (PPARa agonist)	DAT-Cre	F-R curve	No effect	[82]
Pioglitazone (PPARg agonist)	DAT-Cre	F-R curve	↓ oICSS, Shift the F-R curve downward	[82]
Cannabinoid receptor antagonists				
PIMSR (Neutral CB1 antagonist)	DAT-Cre	F-R curve	↓ oICSS, Shift the F-R curve downward	[18]
SR144528 (CB2 antagonist)	DAT-Cre	F-R curve	No effect on oICSS	[83]
GW6471 (PPARa antagonist)	DAT-Cre	F-R curve	↓ oICSS, Shift the F-R curve downward	[82]
GW9662 (PPARg antagonist)	DAT-Cre	F-R curve	No effect on oICSS	[82]

Notes: F-R, stimulation frequency-rate response curve; ↑ indicates an increase; ↓ indicates a reduction.

**Table 3 ijms-25-03455-t003:** Comparisons of the advantages and disadvantages of i.v. self-administration, CPP/CPA, and electrical or optical ICSS approaches in studying drug reward and addiction in rodents.

	Self-Administration	CPP/CPA	ICSS
** Reward assessment **	Measures the reinforcing properties of drugs	Directly measures reward-associative learning Indirectly measures drug reward	Measures the rewarding properties of direct brain stimulation
** Advantages **	Mimics human drug-taking Measures motivation for drugs Allows for the assessment of drug intake patterns over time	Non-invasive Easy to implement	Brain region or cell type specificity Isolates reward circuits Provides real-time measurement of reward-related behavior
** Disadvantages **	Invasive procedure Does not provide information regarding brain regions or neural circuitry involved in drug reward	Does not provide information regarding brain regions or neural circuitry involved in this behavior	Invasive procedure Surgery, AAV, and electrical stimulation may cause brain tissue damage Technically more complicated

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
