# Peer review of "Optical Intracranial Self-Stimulation (oICSS): A New Behavioral Model for Studying Drug Reward and Aversion in Rodents"

_ijms, 2024, doi:10.3390/ijms25063455_

Round 1

Reviewer 1 Report

Comments and Suggestions for Authors

This manuscript explores the contemporary insights into utilizing optical intracranial self-stimulation (oICSS) to understand drug abuse and aversion. Initially, the authors provide a overview of oICSS principles, followed by an in-depth analysis of studies concerning DA-, glutamate-, and GABA-dependent oICSS effects on drug abuse and aversion.

Major Concerns: While oICSS offers neuronal specificity, its utility in advancing the understanding of drug abuse and aversion is limited. The authors should offer a more comprehensive discussion on the constraints associated with employing oICSS.

With oICSS, how do neurotransmitter levels (such as DA, glutamate, GABA) change? Does it accurately replicate the effects of drug abuse or aversion? The intense stimulation/inhibition it induces disrupts brain circuitry, inevitably impacting behavior significantly.

Given the multifaceted mechanisms occurring at molecular, cellular, and circuit levels with drugs of abuse and aversion, how does oICSS influence these processes?

The authors predominantly focus on midbrain oICSS concerning drug abuse and aversion. However, since drug abuse and aversion induce widespread changes across the brain, how applicable is oICSS to other brain regions?

Minor Issue: The title inaccurately implies that oICSS is a novel behavioral model. It has evolved over years and is not a recent development. The authors should consider rephrasing it for accuracy.

Comments on the Quality of English Language

NA

Author Response

This manuscript explores the contemporary insights into utilizing optical intracranial self-stimulation (oICSS) to understand drug abuse and aversion. Initially, the authors provide a overview of oICSS principles, followed by an in-depth analysis of studies concerning DA-, glutamate-, and GABA-dependent oICSS effects on drug abuse and aversion.

Response: We thank the reviewer for his/her understanding of this review article.

Major Concerns: While oICSS offers neuronal specificity, its utility in advancing the understanding of drug abuse and aversion is limited. The authors should offer a more comprehensive discussion on the constraints associated with employing oICSS.

Response: We appreciate the reviewer's thoughtful comment, although we find it disappointing and respectfully disagree. In response to this concern, we have expanded our discussion to thoroughly examine the constraints associated with employing oICSS. Additionally, we have added two paragraphs and a table (Table 3) to compare the advantages and disadvantages of electrical ICSS versus optical ICSS, as well as between ICSS, CPP, and SA, in studying drug reward and addiction. These additions, located at the end of this article (pages 21-23), provide a comprehensive overview of the topic.

Furthermore, we have explored potential strategies and complementary approaches to address these limitations, offering a balanced perspective on the implications of oICSS in studying drug reward and aversion in both humans and experimental animals. We believe these enhancements significantly improve the quality and relevance of our review article.

With oICSS, how do neurotransmitter levels (such as DA, glutamate, GABA) change? Does it accurately replicate the effects of drug abuse or aversion? The intense stimulation/inhibition it induces disrupts brain circuitry, inevitably impacting behavior significantly.

Response: These are indeed excellent comments. The modulation of neurotransmitter levels, including DA, glutamate, and GABA, in response to oICSS stimulation is a complex and multifaceted phenomenon that requires careful consideration. While oICSS can induce changes in neurotransmitter release within targeted brain regions, the extent and specificity of these changes may vary depending on several factors, including the parameters of stimulation (e.g., intensity, frequency, duration), the specific neural populations targeted, and the experimental conditions.

It is important to note that while oICSS can provide valuable insights into the neural circuits underlying reward and aversion, it may not fully replicate the effects of drug abuse or aversion observed in preclinical and clinical settings. The intense stimulation or inhibition induced by oICSS may disrupt brain circuitry and significantly impact behavior, potentially leading to non-physiological responses or alterations in neurotransmitter dynamics that do not accurately reflect the complexities of drug-induced behaviors.

Furthermore, the interpretation of oICSS findings in the context of drug abuse or aversion requires careful consideration of the limitations and potential confounding factors associated with this technique. These may include long-term effects and compensatory responses that may arise in response to chronic or repeated stimulation.

In the revision of this review, we have provided a thorough discussion of these considerations and acknowledge the limitations of oICSS in accurately replicating the effects of drug abuse or aversion (on page 20). We thank the reviewer again for raising these important points, which undoubtedly enrich the quality and relevance of our review article.

Given the multifaceted mechanisms occurring at molecular, cellular, and circuit levels with drugs of abuse and aversion, how does oICSS influence these processes?

Response: This is another excellent comment. We appreciate the opportunity to delve into this complex issue and provide a nuanced understanding of the interactions between oICSS and these processes.

Given the multifaceted mechanisms occurring at molecular, cellular, and circuit levels with drugs of abuse and aversion, oICSS influences these processes in several ways.

At the molecular level, oICSS can modulate neurotransmitter release, receptor activation, and intracellular signaling pathways within targeted brain regions. For instance, oICSS-induced activation of dopaminergic neurons in the mesolimbic pathway may lead to changes in dopamine release and downstream signaling cascades implicated in reward processing.

At the cellular level, oICSS can alter the excitability and firing patterns of specific neuronal populations, resulting in changes in synaptic strength, neuronal plasticity, and network activity. This may involve the activation or inhibition of distinct cell types within neural circuits associated with drug reward and aversion, influencing their overall functional dynamics.

At the circuit level, oICSS can selectively modulate the activity of interconnected brain regions involved in the processing of reward-related stimuli and the regulation of motivated behavior. By targeting specific neural circuits implicated in drug reward and aversion pathways, oICSS enables researchers to dissect the functional connectivity and causal relationships underlying these complex behaviors.

It is important to recognize that while oICSS provides valuable insights into the neural mechanisms underlying drug reward and aversion, it represents a simplified experimental paradigm that may not fully capture the intricacies of these processes in vivo. The effects of oICSS on molecular, cellular, and circuit-level processes must be interpreted within the context of its experimental limitations and potential confounding factors.

In the revised manuscript (p. 19-20), we have provided a thorough discussion of how oICSS influences these processes and the implications for understanding the neurobiology of addiction and aversion.

The authors predominantly focus on midbrain oICSS concerning drug abuse and aversion. However, since drug abuse and aversion induce widespread changes across the brain, how applicable is oICSS to other brain regions?

Response:  Thank you for your insightful observation regarding the predominantly focused on midbrain optogenetic intracranial self-stimulation (oICSS) in our review article concerning drug abuse and aversion. We appreciate the opportunity to address the broader applicability of oICSS to other brain regions implicated in these complex behaviors.

While our review primarily highlights studies utilizing midbrain oICSS to investigate drug reward and aversion, we acknowledge that drug abuse and aversion induce widespread changes across the brain, involving multiple interconnected neural circuits and neurotransmitter systems. As such, the applicability of oICSS to other brain regions beyond the midbrain warrants careful consideration. Indeed, oICSS techniques can be applied to various brain regions beyond the midbrain, including but not limited to the prefrontal cortex, nucleus accumbens, amygdala, lateral habenula, and hippocampus (as we highlighted in Table 1), which play crucial roles in mediating drug-induced behaviors and affective states. By targeting these regions with oICSS, researchers can elucidate the contributions of specific neural circuits to drug reward and aversion and gain insights into the distributed nature of these processes.

Minor Issue: The title inaccurately implies that oICSS is a novel behavioral model. It has evolved over years and is not a recent development. The authors should consider rephrasing it for accuracy.

Response: We thank the reviewer for raising this issue regarding the accuracy of the title. It seems that the reviewer may have overlooked a key point highlighted in our article: that we were the first (in 2017) to introduce an optical stimulation frequency-lever response functional curve into the field of drug abuse and addiction to evaluates the effects of drugs of abuse and other test drugs on brain-reward function.

While oICSS has evolved over the years and is not a recent development, we did not claim that optical ICSS is a novel technique that has been used exclusively to identify the neural substrates underlying brain reward function. In our article, we highlighted that, in addition to this implication, oICSS, by using an optical stimulation frequency-lever response curve, can be used as a behavioral model or measurement to evaluate the rewarding versus aversive effects of drugs of abuse and many other testing drugs. Based on these considerations, we would like to retain the current title used in our initial submission and seek the reviewer's understanding on this point.

Reviewer 2 Report

Comments and Suggestions for Authors

The article entitled "Optical intracranial self-stimulation (oICSS): A new behavioural model for studying drug reward and aversion" analyses the scientific literature dealing with the phenomenon of reward and aversion. It is therefore a review article that should be mentioned in the title. The articles concern experiments in rats, which is not mentioned, particularly in the abstract.

I have no comment on the substance of the article, which is detailed and well documented.

I find the article poorly organized. Chapter 6 "Advantages and limitations of oICSS" is the most interesting for a non-specialist reader. The purpose of the article is clear. I think it would be much better placed in the introduction.

There are numerous abbreviations, many of which are not defined. It would be useful to have an index summarizing all the abbreviations to make the article easier to read. Otherwise, you must keep going back to find the definition.

The figures are very interesting because they illustrate the text well. On the other hand, the tables are not very informative, especially as there are no comments or summaries in their captions.

A discussion chapter is missing. We understand that the experimental model could be used to test drugs before bringing them to market. There are no proposals for human applications. Could deep brain stimulation techniques be used to treat certain addictions? What is the potential role of techniques that are supposed to act on the nucleus accumbens and the VTA, such as music therapy (1), or on the Locus coeruleus, such as occipital nerve stimulation (2)?

1) Menon V, Levitin DJ. The rewards of music listening: response and physiological connectivity of the mesolimbic system. Neuroimage. 2005 Oct 15;28(1):175-84. doi: 10.1016/j.neuroimage.2005.05.053. Epub 2005 Jul 14. PMID: 16023376.

2) De Ridder D, Vanneste S. Occipital Nerve Field Transcranial Direct Current Stimulation Normalizes Imbalance Between Pain Detecting and Pain Inhibitory Pathways in Fibromyalgia. Neurotherapeutics. 2017 Apr;14(2):484-501. doi: 10.1007/s13311-016-0493-8. PMID: 28004273; PMCID: PMC5398977.

Author Response

The article entitled "Optical intracranial self-stimulation (oICSS): A new behavioural model for studying drug reward and aversion" analyses the scientific literature dealing with the phenomenon of reward and aversion. It is therefore a review article that should be mentioned in the title. The articles concern experiments in rats, which is not mentioned, particularly in the abstract.

I have no comment on the substance of the article, which is detailed and well documented.

Response: We thank the reviewer for such positive comments on this article.

I find the article poorly organized. Chapter 6 "Advantages and limitations of oICSS" is the most interesting for a non-specialist reader. The purpose of the article is clear. I think it would be much better placed in the introduction.

Response: We appreciate the reviewer's concerns regarding the advantages and disadvantages of oICSS techniques. To address this, we have made several revisions to the manuscript. Firstly, we have provided an overview of the advantages of oICSS in the introduction (pages 7-8) to help orient readers and set the stage for the discussion. Additionally, we have included a detailed discussion about the advantages and limitations of oICSS at the end of the article to provide a comprehensive understanding of the technique's utility and constraints.

There are numerous abbreviations, many of which are not defined. It would be useful to have an index summarizing all the abbreviations to make the article easier to read. Otherwise, you must keep going back to find the definition.

Response: We have now provided an “Abbreviation list” at the beginning of the revised manuscript to address the reviewer’s concerns about it.

The figures are very interesting because they illustrate the text well. On the other hand, the tables are not very informative, especially as there are no comments or summaries in their captions.

Response: Thank you for your feedback regarding the figures and tables in our article. We are pleased to hear that the figures have effectively illustrated the text. Regarding the tables, we have revised them and incorporated multiple columns to highlight the major findings in literature reports. These include the AAV targeted brain region and phenotypes of neurons, opsin types (CheR2, Halo, Arch3), stimulation frequency used in each study, and the major behavioral readout – producing oICSS behavior.

A discussion chapter is missing. We understand that the experimental model could be used to test drugs before bringing them to market. There are no proposals for human applications. Could deep brain stimulation techniques be used to treat certain addictions? What is the potential role of techniques that are supposed to act on the nucleus accumbens and the VTA, such as music therapy (1), or on the Locus coeruleus, such as occipital nerve stimulation (2)?

1) Menon V, Levitin DJ. The rewards of music listening: response and physiological connectivity of the mesolimbic system. Neuroimage. 2005 Oct 15;28(1):175-84. doi: 10.1016/j.neuroimage.2005.05.053. Epub 2005 Jul 14. PMID: 16023376.

2) De Ridder D, Vanneste S. Occipital Nerve Field Transcranial Direct Current Stimulation Normalizes Imbalance Between Pain Detecting and Pain Inhibitory Pathways in Fibromyalgia. Neurotherapeutics. 2017 Apr;14(2):484-501. doi: 10.1007/s13311-016-0493-8. PMID: 28004273; PMCID: PMC5398977.

Response: While our review primarily focuses on the implications of oICSS in preclinical research for studying drug reward and aversion mechanisms, we recognize the relevance of considering its potential applications in human contexts, including therapeutic interventions for addiction. Deep brain stimulation (DBS) techniques targeting specific brain regions implicated in addiction, such as the nucleus accumbens and the ventral tegmental area (VTA), have shown promise in preclinical and clinical studies for reducing drug cravings and promoting abstinence. Accordingly, we have added a section titled "Potential Implications of oICSS in Human Studies" (pages 22-23) to address the reviewer's concerns about this topic.

We acknowledge that music has been shown to modulate reward-related brain circuits and promote emotional regulation, potentially reducing the reinforcing effects of drugs and cravings. However, this topic is beyond the scope of this article and will not be discussed further.
